# Scattering Networks on the Sphere for Scalable and Rotationally Equivariant Spherical CNNs

**Jason D. McEwen**,[*] **Christopher G. R. Wallis & Augustine N. Mavor-Parker**
Kagenova Limited, Guildford, Surrey, United Kingdom.

## Abstract

Convolutional neural networks (CNNs) constructed natively on the sphere have been developed recently and shown to be highly effective for the analysis of spherical data. While an efficient framework has been formulated, spherical CNNs are nevertheless highly computationally demanding; typically they cannot scale beyond spherical signals of thousands of pixels. We develop scattering networks constructed natively on the sphere that provide a powerful representational space for spherical data. Spherical scattering networks are computationally scalable and exhibit rotational equivariance, while their representational space is invariant to isometries and provides efficient and stable signal representations. By integrating scattering networks as an additional type of layer in the generalized spherical CNN framework, we show how they can be leveraged to scale spherical CNNs to the high-resolution data typical of many practical applications, with spherical signals of many tens of megapixels and beyond.

## 1 Introduction

The construction of convolutional neural networks (CNNs) on the sphere has received considerable attention recently, opening up their application to fields where standard (Euclidean) CNNs are not effective. Such techniques are finding application in diverse fields; for example, for the analysis of 360° photos and videos in virtual reality and for the analysis of astrophysical observations on the celestial sphere in cosmology. In both of these fields high-resolution spherical images are typically encountered, with image resolutions often reaching many tens of megapixels. Current spherical CNN approaches that are defined natively on the sphere and satisfy rotational equivariance, however, are limited to low resolutions and often cannot scale beyond thousands of pixels, severely restricting their applicability. In this article we present a strategy to scale spherical CNNs to high resolution, further opening up their applicability to high-resolution spherical images encountered in virtual reality, cosmology, and many other fields beyond.

A number of spherical CNN constructions have been proposed recently, which can be broadly categorized into two approaches: (i) those that are defined natively on the sphere and capture rotational equivariance (e.g. Cohen et al., 2018a; Esteves et al., 2018; 2020; Kondor et al., 2018; Cobb et al., 2021); and (ii) those that are constructed on discretizations of the sphere and typically do not fully capture rotational equivariance (e.g. Boomsma & Frellsen, 2017; Jiang et al., 2018; Perraudin et al., 2019; Cohen et al., 2019). While constructions on discretized representations of the sphere can often be computed efficiently, and some capture rotational equivariance to a small set of rotations (Cohen et al., 2019), such approaches necessitate an approximate representation of spherical signals, losing the connection to the underlying continuous symmetries of the sphere and thus cannot fully capture rotational equivariance. Approaches constructed natively on the sphere that do capture rotational equivariance, on the other hand, are highly computationally demanding. Despite considerable improvements in computational efficiency proposed recently (Cobb et al., 2021), such approaches nevertheless remain computationally demanding and are not scalable to high resolution.

We seek a framework that captures the underlying continuous symmetries of the sphere and exhibits rotational equivariance, while also being computationally scalable to high resolution. We draw

---

[*]Corresponding author: jason.mcewen@kagenova.com

inspiration from Cobb et al. (2021), which presents a generalized spherical CNN framework and advocates hybrid networks where different types of spherical CNN layers are leveraged alongside each other. We consider an additional layer to be integrated into this hybrid approach that must, critically, be scalable to high resolution and allow subsequent layers to operate effectively at low resolution, while also exhibiting rotational equivariance. To provide an effective representational space it should also provide efficient and stable signal representations, with control of the scale of signal invariances, and be sensitive to all signal content. Scattering networks meet all requirements.

Scattering networks were first proposed in the seminal work of Mallat (2012) in an effort to provide some further theoretical basis explaining the observed empirical effectiveness of CNNs (elaborated in Mallat 2016). They are constructed from a cascade of linear transforms (typically convolutions), combined with pointwise non-linear activation functions, with carefully designed filters to ensure certain stability and invariance properties. A scattering network is thus effectively a CNN but with designed, rather than learned, filters. Typically wavelets that themselves satisfy certain stability and invariance properties are adopted for the filters and the absolute value function is adopted for the non-linear activation function. Scattering networks were initially constructed on Euclidean domains and have been applied in a variety of applications (e.g. Bruna & Mallat, 2011; 2013; Andén et al., 2019). More recently scattering networks that support non-Euclidean data have been constructed through graph representations (Gama et al., 2019b;a; Gao et al., 2019; Zou & Lerman, 2020; Perlmutter et al., 2019) or on general Riemannian manifolds (Perlmutter et al., 2020). While these approaches are flexible, this flexibility comes at a cost. Just as there are considerable advantages in constructing CNNs natively on the sphere (rather than, e.g., adopting graph CNNs), so too there are advantages in constructing scattering networks natively on the sphere. For example, spherical scattering networks can be constructed to retain a connection to the underlying continuous symmetries of the sphere and to leverage fast algorithms tailored to the harmonic structure of the sphere.

In this article we construct scattering networks on the sphere, motivated by their use as an initial layer in hybrid spherical CNNs in order to yield rotationally equivariant spherical CNNs that scale to high resolution. In Section 2 we review spherical signal representations and describe the invariance and stability properties required to provide an effective representational space. In Section 3 we construct scattering networks on the sphere and discuss their invariance and stability properties. In Section 4 we integrate scattering networks into the existing generalized spherical CNN framework. Experiments studying the properties of spherical scattering networks and demonstrating their effectiveness in hybrid spherical CNNs are shown in Section 5. Concluding remarks are made in Section 6.

## 2 SPHERICAL SIGNAL REPRESENTATIONS

We seek a representational space that can be used as an initial layer in hybrid spherical CNNs to scale to high-resolution data. Such a representational space must, critically, be scalable, allow subsequent CNN layers to operate at low resolution, and be rotationally equivariant. Moreover, to provide an effective representation for machine learning problems the space should provide efficient and stable signal representations, with control over the scale of signal invariances, while being sensitive to all signal content. In this section we concisely review common spherical signal representational spaces, including spatial, harmonic and wavelet signal representations, before discussing the desired invariance and stability properties of the representational space that we seek.

### 2.1 SPATIAL AND HARMONIC REPRESENTATIONS

Consider square integrable signals $f, g \in L^2(\mathbb{S}^2)$ on the sphere $\mathbb{S}^2$ with inner product $\langle f, g \rangle$. The inner product induces the norm $\|f\|_2 = \sqrt{\langle f, f \rangle}$, with metric $d_{L^2(\mathbb{S}^2)}(f, g) = \|f - g\|_2$. We consider spherical signals bandlimited at $L$, with harmonic coefficients $\hat{f}_{\ell m} = \langle f, Y_{\ell m} \rangle = 0, \forall \ell \geq L$, where $Y_{\ell m}$ are the spherical harmonic functions of natural degree $\ell$ and integer order $|m| \leq \ell$. Sampling theories on the sphere (Driscoll & Healy, 1994; McEwen & Wiaux, 2011a) provide a mechanism to capture all information content of an underlying continuous function from a finite set of samples. By providing access to the underlying continuous signal, such a representation perfectly captures all spherical symmetries and geometric properties. We adopt the sampling theorem on the sphere of McEwen & Wiaux (2011a) since it provides the most efficient sampled signal representation (reducing the Nyquist rate by a factor of two compared to Driscoll & Healy 1994).

## 2.2 WAVELET REPRESENTATIONS

A number of wavelet frameworks on the sphere have been constructed (see Appendix A). We focus on spherical scale-discretized wavelets since they have an underlying continuous representation, exhibit excellent localization and asymptotic correlation properties, are a Parseval frame, support directional wavelets, and exhibit fast algorithms for exact and efficient computation (McEwen et al., 2007; Wiaux et al., 2008; Leistedt et al., 2013; McEwen et al., 2013; 2015c; 2018). Furthermore, the wavelet transform is rotationally equivariant in theory (since it is based on spherical convolutions) and in practice (since its computation leverages underlying spherical sampling theory).

The (axisymmetric) spherical scale-discretized wavelet transform is defined by the convolution of $f \in \mathrm{L}^2(\mathbb{S}^2)$ with the *wavelet* $\psi_j \in \mathrm{L}^2(\mathbb{S}^2)$, yielding *wavelet coefficients* $W_j(\omega) = (f \star \psi_j)(\omega) = \langle f, R_\omega \psi_j \rangle$, where $\omega = (\theta, \varphi)$ denotes a point on the sphere with colatitude $\theta \in [0, \pi]$ and longitude $\varphi \in [0, 2\pi)$, and $R_\omega = R_{(\varphi, \theta, 0)}$ denotes a three-dimensional (3D) rotation (we adopt the active $zyz$ Euler convention). The wavelet scale $j$ encodes the angular localization of $\psi_j$, with increasing $j$ corresponding to smaller scale (i.e. higher frequency) signal content. The minimum and maximum wavelet scales considered are denoted $J_0$ and $J$, i.e. $0 \leq J_0 \leq j \leq J$. The wavelet coefficients capture high-frequency or bandpass information of the underlying signal; they do not capture low-frequency signal content. A *scaling function* $\phi \in \mathrm{L}^2(\mathbb{S}^2)$ is introduced to capture low-frequency signal content, with *scaling coefficients* given by $W(\omega) = (f \star \phi)(\omega) = \langle f, R_\omega \phi \rangle$. For notational convenience we introduce the operator notation $w_j = \Psi_j f = f \star \psi_j$ and $w = \Phi f = f \star \phi$.

Further details regarding the spherical scale-discretized wavelet construction, synthesis and properties are contained in Appendix A. Note that the wavelet construction relies on a fixed dilation parameter $\lambda \in \mathbb{R}_*^+, \lambda > 1$, that controls the relative scaling of harmonic degrees one wishes each wavelet to probe. A common choice is dyadic scaling with $\lambda = 2$. The minimum wavelet scale $J_0$ may be chosen freely or may be set by specifying a desired bandlimit $L_0$ for the scaling coefficients, yielding $J_0 = \lceil \log_\lambda L_0 \rceil$. For $J_0 = 0$ the wavelets probe the entire frequency content of the signal of interest except for its mean, which is encoded in the scaling coefficients. The wavelet transform of a signal bandlimited at $L$ is therefore specified by the parameters $(L, \lambda, J_0)$, or alternatively $(L, \lambda, L_0)$.

## 2.3 ISOMETRIES AND DIFFEOMORPHISMS

While we have mentioned the signal representation we seek must satisfy invariance and stability properties, we are yet to elaborate on these properties in detail. We define the transformations under which these properties are sought, before describing the properties themselves in the next subsection.

Consider transforms of spherical signals that are an *isometry*, preserving the distance between signals. More precisely, consider the isometry $\zeta \in \mathrm{Isom}(\mathbb{S}^2)$, where $\mathrm{Isom}(\mathbb{S}^2)$ denotes the isometry group of the sphere. The action of an isometric transformation $V_\zeta : \mathrm{L}^2(\mathbb{S}^2) \to \mathrm{L}^2(\mathbb{S}^2)$ is defined as $V_\zeta f(\omega) = f(\zeta^{-1}\omega)$. The distance between signals $f, g \in \mathrm{L}^2(\mathbb{S}^2)$ is then preserved under an isometry, i.e. $d_{\mathrm{L}^2(\mathbb{S}^2)}(V_\zeta f, V_\zeta g) = d_{\mathrm{L}^2(\mathbb{S}^2)}(f, g)$. One of the most typical isometries considered for spherical signals is a 3D rotation $R_\rho$, often parameterized by Euler angles $\rho = (\alpha, \beta, \gamma) \in \mathrm{SO}(3)$.

Consider *deformations* of spherical signals, formally defined by *diffeomorphisms*. A diffeomorphism is a transformation that is a differentiable bijection (a one-to-one mapping that is invertible), with an inverse that is also differentiable. Loosely speaking, a diffeomorphism can be considered as a "well-behaved" deformation. Consider a diffeomorphism $\zeta \in \mathrm{Diff}(\mathbb{S}^2)$, where $\mathrm{Diff}(\mathbb{S}^2)$ denotes the diffeomorphism group of the sphere. The action of a diffeomorphism on a spherical signal is defined similarly to an isometric transformation. It is useful to quantify the size of a diffeomorphism. Let $\|\zeta\|_\infty = \sup_{\omega \in \mathbb{S}^2} d_{\mathbb{S}^2}(\omega, \zeta(\omega))$ represent the maximum displacement between points $\omega$ and $\zeta(\omega)$, where $d_{\mathbb{S}^2}(\omega, \omega')$ is the geodesic distance on the sphere between $\omega$ and $\omega'$. Often we are interested in small diffeomorphisms about an isometry such as a rotation, denoted $\zeta' = \zeta_1 \circ \zeta_2$ for some isometry $\zeta_1 \in \mathrm{Isom}(\mathbb{S}^2)$, e.g. $\rho \in \mathrm{SO}(3)$, and some diffeomorphism $\zeta_2 \in \mathrm{Diff}(\mathbb{S}^2)$.

## 2.4 INVARIANT AND STABLE REPRESENTATIONS

In representational learning, invariances with respect to isometries and stability with respect to diffeomorphisms play crucial roles. As an illustrative example consider the classification of hand-written digits in planar (spherical) images. Translation (rotation) is a common isometry to which we

often seek to encode various degrees of invariance as an inductive bias in machine learning models. In some cases small translations (rotations) may be responsible for intra-class variations (e.g. if the digit is always located close to the center of the image), whereas in others global translations (rotations) may not alter class instances (e.g. if the digit may be located anywhere in the image). Invariance to isometries up to some scale is therefore an important property of effective representation spaces. Intra-class variations may also be due to small diffeomorphisms (e.g. deformations in the way a given digit is written), while inter-class variations may be due to larger diffeomorphisms (e.g. deformations mapping one digit to another). Consequently, an effective representation space is one that is stable to diffeomorphisms, i.e. is insensitive to small diffeomorphisms and changes by an increasing amount as the size of the diffeomorphism increases. It is well-known that scattering networks constructed from cascades of appropriate wavelet representations, combined with non-linear activation functions, satisfy these invariance and stability properties, thus providing an effective representation space (e.g. Mallat, 2012; 2016).

## 3    SCATTERING NETWORKS ON THE SPHERE

We present the construction of scattering networks on the sphere, which follows by a direct analogy with the Euclidean construction of Mallat (2012). Firstly, we describe conceptually how scattering representations satisfy the invariance and stability properties that we seek. Secondly, we define the spherical scattering propagator and transform, constructed using the spherical scale-discretized wavelet transform (e.g. McEwen et al., 2018). Thirdly, we use the scattering transform to define scattering networks. Finally, we formalize and prove that scattering networks are invariant to isometries up to a given scale and are stable to diffeomorphisms.

### 3.1    SCATTERING REPRESENTATIONS

*Scattering representations* leverage stable wavelet representations to create powerful hierarchical representational spaces that satisfy isometric invariance up to a particular scale, are stable to diffeomorphisms and probe all signal content (both low and high frequencies). The scaling coefficients of the scale-discretized wavelet transform on the sphere are computed by an averaging (smoothing) operation. Scaling coefficients therefore satisfy isometric invariance associated with the scale of the scaling function (i.e. $J_0$ or equivalently $L_0$). However, scaling coefficients clearly do not probe all signal content as they are restricted to low frequencies. Wavelet coefficients of the scale-discretized wavelet transform on the sphere do not satisfy isometric invariance. However, they are stable to diffeomorphisms by grouping frequencies into packets through the Meyer-like tiling of the spherical harmonic space on which they are constructed. Furthermore, they probe signal content at a range of frequencies, including high frequency signal content. A wavelet signal representation, including scaling and wavelet coefficients, does *not* recover a representational space with the desired invariance and stability properties: scaling coefficients satisfy some desired properties, whereas wavelet coefficients satisfy others. However, by combining cascades of scaling and wavelet coefficient representations, with non-linear activation functions, it is possible to recover a representation that exhibits all of the desired properties. Scattering representations do precisely this. Furthermore, all signal content is then probed, with high frequency signal content non-linearly mixed into low frequencies (Mallat, 2012).

### 3.2    SCATTERING PROPAGATOR AND TRANSFORM

Scattering networks are constructed from a scattering transform, which itself is constructed using a scattering propagator. The spherical scattering propagator for scale $j$ is defined by

$$U[j]f = A\Psi_j f = |f \star \psi_j|, \tag{1}$$

where a non-linear activation function $A : L^2(\mathbb{S}^2) \to L^2(\mathbb{S}^2)$ is applied to the wavelet coefficients at scale $j$. Following Mallat (2012), we adopt the absolute value (modulus) function, $Af = |f|$, for the activation function since it is non-expansive and thus preserves the stability of wavelet representations. Moreover, as it acts on spherical signals pointwise, it is rotationally equivariant. Scattering propagators can then be constructed by applying a cascade of propagators:

$$U[p]f = U[j_1, j_2, \ldots, j_d]f = U[j_d] \ldots U[j_2]U[j_1]f = |||f \star \psi_{j_1}| \star \psi_{j_2}| \ldots \star \psi_{j_d}|, \tag{2}$$

for the path $p = (j_1, j_2, \ldots, j_d)$ with depth $d$. By adopting carefully designed filters, i.e. wavelets, combined with a non-expansive activation function, scattering propagators inherit stability properties

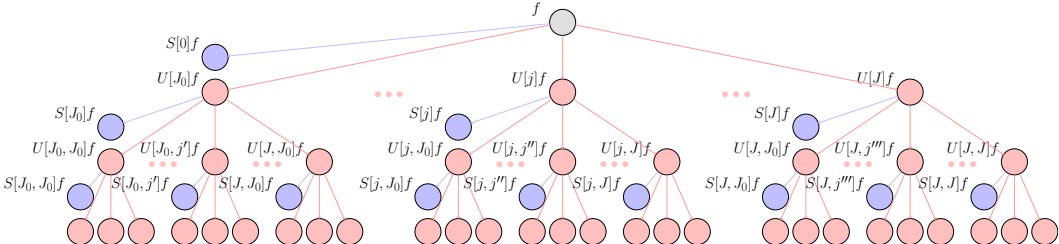

Figure 1: Spherical scattering network of the signal $f \in \mathrm{L}^2(\mathbb{S}^2)$. The signal is propagated through cascades of spherical scale-discretized wavelet transforms, combined with absolute value activation functions, i.e. $U[p]f = |||f \star \psi_{j_1}| \star \psi_{j_2}| \ldots \star \psi_{j_d}|$, denoted by red nodes. The outputs of the scattering network are given by projecting these signals onto the spherical wavelet scaling function, resulting in scattering coefficients $S[p]f = |||f \star \psi_{j_1}| \star \psi_{j_2}| \ldots \star \psi_{j_d}| \star \phi$, denoted by blue nodes.

from the underlying wavelets, yielding representations that are stable to diffeomorphisms. Scattering propagators, however, do not yield representations that exhibit isometric invariances. In order to recover a representation that does, and to control the scale of invariance, the *scattering transform* is constructed by projection onto the scaling function, yielding *scattering coefficients*

$$S[p]f = \Phi U[p]f = |||f \star \psi_{j_1}| \star \psi_{j_2}| \ldots \star \psi_{j_d}| \star \phi, \tag{3}$$

for path $p$. We adopt the convention that $S[p = 0]f = \Phi f$. By projecting onto the wavelet scaling function we inherit the isometric invariance of the scaling coefficients, with the scale of invariance controlled by the scale of the scaling function (i.e. by the $J_0$ or $L_0$ parameter). Consequently, the scattering transform yields a representational space that satisfies both the desired invariance and stability properties (shown formally in Section 3.4). Furthermore, since scattering representations are constructed from wavelet transforms and pointwise non-linear activation functions, both of which are rotationally equivariant, the resulting scattered representation is also rotationally equivariant. For $J_0 = 0$ the scaling function is constant and we recover a representation that is completely invariant to isometries (in many settings we do not desire full invariance to rotations and so we set $J_0 > 0$ in the numerical experiments that follow). Formally, in the case where $J_0 = 0$ and the network is fully rotationally invariant, equivariance still holds, it is just that the spherical scattering coefficients for each path are reduced to a single constant value over the sphere. Note also that the computation of the spherical scattering transform is scalable since it is based on spherical scale-discretized wavelet transforms that are themselves computationally scalable (McEwen et al., 2007; 2013; 2015c).

## 3.3 SCATTERING NETWORKS

A *scattering network* is then simply a collection of scattering transforms for a number of paths: $\mathcal{S}_{\mathbb{P}}f = \{S[p]f : p \in \mathbb{P}\}$, where the general path set $\mathbb{P}$ denotes the infinite set of all possible paths $\mathbb{P} = \{p = (j_1, j_2, \ldots, j_d) : J_0 \leq j_i \leq J, 1 \leq i \leq d, d \in \mathbb{N}_0\}$. Since a scattering network is a collection of scattering transforms that are rotationally equivariant, so too are scattering networks. In practice one typically considers a path set defined by paths up to some maximum depth $D$, i.e. $\mathbb{P}_D = \{p = (j_1, j_2, \ldots, j_d) : J_0 \leq j_i \leq J, 1 \leq i \leq d, 0 \leq d \leq D\}$. A diagram of a scattering network is illustrated in Fig. 1. A scattering network is effectively a spherical CNN with carefully designed filters (i.e. wavelets) and typically with an activation function given by the absolute value function. Since the intended use of spherical scattering networks here is as an initial layer in generalized spherical CNNs, where we seek to mix information from high to low frequencies so that subsequent CNN layers can operate at low resolution, we are often interested in *descending paths* where the wavelet scale $j$ reduces along the path. The descending path set, up to maximum depth $D$, is given by $\mathbb{P}_D^{\mathrm{descending}} = \{p = (j_1, j_2, \ldots, j_d) : J_0 \leq j_i \leq J, j_1 \geq \ldots \geq j_d, 1 \leq i \leq d, 0 \leq d \leq D\}$. Since scattering networks are simply a collection of scattering transforms, the network inherits the properties of scattering transforms discussed previously and thus satisfies all requirements of the representational space that we seek.

## 3.4 ISOMETRIC INVARIANCE AND STABILITY TO DIFFEOMORPHISMS

While we have explained conceptually how spherical scattering networks exhibit isometric invariance and are stable to diffeomorphisms, here we formalize these results.

**Theorem 1** (**Isometric Invariance**). *Let $\zeta \in \mathrm{Isom}(\mathbb{S}^2)$, then there exists a constant $C$ such that for all $f \in \mathrm{L}^2(\mathbb{S}^2)$,*

$$\|\mathcal{S}_{\mathbb{P}_D} f - \mathcal{S}_{\mathbb{P}_D} V_\zeta f\|_2 \le CL^{5/2}(D+1)^{1/2}\lambda^{J_0}\|\zeta\|_\infty\|f\|_2. \tag{4}$$

Theorem 1 is analogous to Theorem 3.2 of Perlmutter et al. (2020) that considers scattering networks on general Riemannian manifolds. The manifold wavelet construction considered in Perlmutter et al. (2020) is analogous to the spherical scale-discretized wavelet construction (e.g. McEwen et al., 2018) since both are based on a Meyer-like tiling of the harmonic frequency line. However, in each approach different wavelet generating functions are considered, with an exponential kernel adopted in Perlmutter et al. (2020) and an infinitely differential Schwartz kernel adopted for spherical scale-discretized wavelets. Consequently, the proof of Theorem 1 follows directly from the proof presented in Perlmutter et al. (2020) but accounting for different wavelet generating functions and noting that $\sum_\ell \hat{\phi}_{\ell 0} \le L^{1/2}\lambda^{J_0}$. Theorem 1 shows that the scattering network representation is invariant to isometries up to the scale controlled by $J_0$, which is directly related to the bandlimit $L_0$ of the spherical wavelet scaling function.

**Theorem 2** (**Stability to Diffeomorphisms**). *Let $\zeta \in \mathrm{Diff}(\mathbb{S}^2)$. If $\zeta = \zeta_1 \circ \zeta_2$ for some isometry $\zeta_1 \in \mathrm{Isom}(\mathbb{S}^2)$ and diffeomorphism $\zeta_2 \in \mathrm{Diff}(\mathbb{S}^2)$, then there exists a constant $C$ such that for all $f \in \mathrm{L}^2(\mathbb{S}^2)$,*

$$\|\mathcal{S}_{\mathbb{P}_D} f - \mathcal{S}_{\mathbb{P}_D} V_\zeta f\|_2 \le CL^2 \times \left[ L^{1/2}(D+1)^{1/2}\lambda^{J_0}\|\zeta_1\|_\infty + L^2\|\zeta_2\|_\infty \right]\|f\|_2. \tag{5}$$

Theorem 2 is analogous to Theorem 4.1 of Perlmutter et al. (2020) and the proof again follows similarly. Theorem 2 shows that the scattering network representation is stable to small diffeomorphisms $\zeta_2$ about an isometry. Moreover the change in signal representation depends on the size of the diffeomorphism $\|\zeta_2\|_\infty$. Consequently, for a classification problem, for example, the scattering network representation is likely to be relatively insensitive to intra-class variations due to small diffeomorphisms but sensitive to inter-class variations due to large diffeomorphisms.

## 4 Leveraging Scattering Networks for Scalable Spherical CNNs

We describe how spherical scattering networks may be integrated into the generalized spherical CNN framework of Cobb et al. (2021) to scale to high-resolution data in a rotationally equivariant manner.

### 4.1 Generalized Spherical CNNs

A variety of spherical CNN constructions have been presented recently in the influential works of Cohen et al. (2018a), Esteves et al. (2018) and Kondor et al. (2018). These were unified in a generalized framework by Cobb et al. (2021), who advocate hybrid networks where varying types of spherical layers can be leveraged alongside each other, and where alternative techniques to substantially improve efficiency were also developed.

Following Cobb et al. (2021), consider the generalized spherical CNN framework where the $s$-th layer of the network takes the form of a triple $\mathcal{A}^{(s)} = (\mathcal{L}_1, \mathcal{N}, \mathcal{L}_2)$, comprised of linear, non-linear, and linear operators, respectively. The output of layer $s$ is then given by $f^{(s)} = \mathcal{A}^{(s)}(f^{(s-1)}) = (\mathcal{L}_1 \circ \mathcal{N} \circ \mathcal{L}_2)(f^{(s-1)})$, where $f^{(s-1)}$ is the signal inputted to the layer. The linear and non-linear operators can take a variety of forms, such as those introduced in Cohen et al. (2018a); Esteves et al. (2018); Kondor et al. (2018); Cobb et al. (2021). The linear layers are typically convolutions, either on the sphere, rotation group, or generalized convolutions (Kondor et al., 2018; Cobb et al., 2021), while the non-linear layers are pointwise or tensor-product activations based on Clebsch-Gordan decompositions (Kondor et al., 2018; Cobb et al., 2021). While generalized spherical CNNs have been shown to be highly effective and techniques to compute them efficiently have been developed (Cobb et al., 2021), they remain highly computationally demanding and cannot scale to high-resolution data.

### 4.2 Scalable Generalized Spherical CNNs

Spherical scattering networks (Section 3) can be leveraged to scale spherical CNNs to high resolution, whilst maintaining rotational equivariance. Adopting the generalized spherical CNN framework of Cobb et al. (2021), we advocate the use of scattering networks as an additional type of layer.

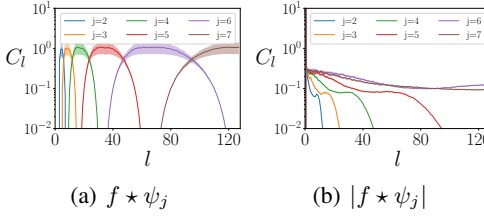

(a) $f \star \psi_j$  (b) $|f \star \psi_j|$

Figure 2: Power spectra of wavelet coefficients, for $(L, \lambda, J_0) = (128, 2, 2)$, before and after application of the absolute value activation function (mean solid; standard deviation shaded).

Table 1: Equivariance error and median relative energy of scattering coefficients of varying path depth $d$, for $(L, \lambda, L_0) = (256, 2, 32)$.

| Path depth $d$ | Equivariance Error | | | Energy |
|---|---|---|---|---|
| | Minimum | Median | Maximum | |
| 0 | – | 0.00% | – | 9.41% |
| 1 | 0.01% | 0.05% | 0.24% | 15.56% |
| 2 | 0.18% | 1.01% | 5.36% | 1.39% |
| 3 | 0.56% | 3.47% | 10.68% | 0.16% |

Spherical scattering networks are computationally scalable, rotationally equivariant, exhibit isometric invariance up to some scale, provide efficient and stable representations, and are sensitive to all signal content, including both low and high frequencies. Consequently, scattering networks provide a highly effective representational space for machine learning problems. Each set of scattering coefficients, in the parlance of CNNs, simply provides an additional channel.

Computation of the spherical scattering network scales as $\mathcal{O}(L^3)$, for bandlimit $L$, due to the adoption of fast spherical wavelet transforms (McEwen et al., 2007; Leistedt et al., 2013; McEwen et al., 2013; 2015c), which themselves leverage fast harmonic transforms (McEwen & Wiaux, 2011b; McEwen et al., 2015b). In contrast, even the efficient spherical CNNs of Cobb et al. (2021) typically scale as $\mathcal{O}(L^4)$. More critically, however, scattering networks mix signal content from high to low frequencies. Thus, they are an ideal first layer in a hybrid spherical CNN so that subsequent layers may then operate at low resolution, while still capturing high-frequency signal content.

Since scattering networks use designed filters, rather than learned filters, they do not need to be trained. Consequently, a scattering network layer acting as the first layer of a spherical CNN can be considered as a data preprocessing stage before training. The outputs of the scattering network therefore need only be computed once, rather than repeatedly during training. Furthermore, the size of the output data of a scattering network is modest since all sets of output scattering coefficients (i.e. output channels) are limited to the resolution of the scaling function $L_0$ and so need only be stored at low resolution. These features provide considerable computational and memory savings, allowing scattering networks to scale spherical CNNs to high resolutions. For example, the spherical scale-discretized wavelet transform on which the scattering network is based has so far been applied up to resolution $L = 8192$ (Rogers et al., 2016a;b) and higher resolutions are feasible. While evaluation of a scattering network at very high resolution may take some time, this is not a concern since the computation is trivially parallelizable and need only be computed once outside of training.

## 5 EXPERIMENTS

We perform numerical experiments to verify the theoretical properties of spherical scattering networks. Using spherical scattering networks as an initial layer in the generalized spherical CNN framework, we show how scattering networks can be leveraged to scale spherical CNNs to high resolutions. Spherical scattering networks are implemented in the `fourpiAI` [1] software package. We use the `s2let` [2] code to perform spherical wavelet transforms.

### 5.1 POWER OF SCATTERED SIGNALS

The absolute value function adopted as the activation function within scattering networks acts to spread signal content, predominantly from high to low frequencies. To demonstrate this we generate 100 simulations of random spherical signals, with spherical harmonic coefficients drawn from a standard Gaussian distribution. In Fig. 2 we plot the average power spectra of the wavelet coefficients of these signals before and after taking the absolute value. Notice in Fig. 2(a) that the harmonic support of the wavelet coefficients matches the support of the underlying wavelets, as expected. Notice in Fig. 2(b) that after taking the absolute value the power of the signal is spread in harmonic space, predominantly mixed from high to low frequencies, and that variability is reduced, as expected.

---

[1] www.kagenova.com/products/fourpiAI

[2] www.s2let.org

## 5.2 EQUIVARIANCE TESTS

To test the equivariance of the scattering transform we consider 100 random test signals and compute the mean relative equivariance error between rotating signals before and after the scattering transform (see Appendix B.1 for further details). Results are given in Table 1, averaged over all scattering coefficients with the same path depth $d$. Notice that equivariance errors are small, as expected. In fact, equivariance errors are considerably smaller than the standard spherical ReLU layer commonly used in spherical CNNs, which has an error on the order of $\sim 35\%$ (Cobb et al., 2021). Notice also that the signal energy for each channel is small for a path length of three and greater, again as expected.

## 5.3 DIFFEOMORPHISM TESTS

To test the stability of spherical scattering networks to diffeomorphisms we consider the following experiment. Taking the prototypical example of classification of hand-written digits, we compute differences in scattering network representations for both inter- and intra-class instances using MNIST digits projected onto the sphere (Cohen et al., 2018b). See Appendix B.2 for further details. Results are shown in Fig. 3 for all combinations of digits, normalized relative to intra-class variations of the digit one (which, incidentally, exhibits the smallest intra-class variation, as might be expected). As discussed in Section 2.4, small diffeomorphisms are likely responsible for intra-class variations and large diffeomorphisms for inter-class variations. Furthermore, by Theorem 2, a larger diffeomorphism leads to a larger difference in scattering representation. For MNIST digits, we therefore expect the average difference between scattering representations to increase as the size of diffeomorphisms within and between digits

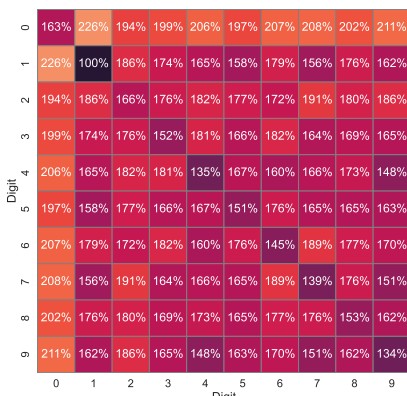

Figure 3: Differences in scattering network representations for spherical MNIST digits, demonstrating stability to diffeomorphisms.

increases. This is indeed what we observe. For each digit, intra-class differences are smaller than inter-class differences, i.e. the diagonal elements of Fig. 3 are smaller than the other elements on the same row/column. While it is difficult to quantify the size of a diffeomorphism mapping one digit to another, the results generally match ones intuition about similarities between digits, e.g. a three is more similar to a seven and nine, than to a zero or four. In general, the findings of this experiment support the theoretical result that spherical scattering networks are stable to diffeomorphisms.

## 5.4 SPHERICAL MNIST AT VARYING RESOLUTION

We consider a variant of the standard benchmark of classifying MNIST digits projected onto the sphere (Cohen et al., 2018b), varying the resolution of spherical images and considering digits projected at smaller angular sizes (see Fig. 4). The intention is to evaluate the effectiveness of scattering networks to represent high-resolution signal content in low-resolution scattering coefficients. To this end, we consider two classification models (see Appendix B.3). The first is the hybrid model proposed in Cobb et al. (2021). Since this model is too computationally demanding to run at high resolution, it considers harmonic content of the spherical MNIST images up to $L = 32$ only. The second model prepends a scattering network to the first model with output scattering coefficients at resolution $L_0 = 32$. We stress that this experiment is constructed specifically to test the effectiveness of the scattering network to spread signal content to low frequencies for subsequent low-frequency spherical CNN layers. The model without a scattering network will not perform well since it loses high-frequency signal content. As resolution is increased, the digit size is reduced so that signal content is concentrated at higher frequencies. The problem thus becomes increasingly more difficult for the models considered and we are thus able to observe how effective the scattering network is at mixing information into low frequencies. We consider two scenarios: digits that are centered and *not* rotated during training and testing (NR/NR); and digits that are *not* rotated during training but rotated to arbitrary positions on the sphere when testing (NR/R). Effective classification for the latter scenario requires invariance of the model. The results of these experiments are shown in Table 2. Since the problem is constructed to be more difficult as resolution increases, we observe accuracy decreasing as resolution increases. In all scenarios classification accuracy is significantly improved when incorporating a scattering

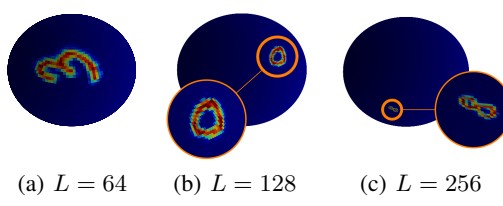

(a) $L = 64$    (b) $L = 128$    (c) $L = 256$

Figure 4: MNIST digits projected onto the sphere at varying resolution.

Table 2: Spherical MNIST performance.

| $L$ | Digit Size | Model | Accuracy NR/NR | NR/R |
|---|---|---|---|---|
| 64 | 82.2° | without scattering | 98.29 | 88.66 |
| 64 | 82.2° | with scattering | 98.80 | 97.22 |
| 128 | 42.5° | without scattering | 92.49 | 51.71 |
| 128 | 42.5° | with scattering | 94.70 | 76.81 |
| 256 | 21.4° | without scattering | 75.12 | 17.23 |
| 256 | 21.4° | with scattering | 96.02 | 59.48 |

network since the scattering network provides an effective representational space that mixes high-frequency signal content to low frequencies, where it may be captured in the following spherical CNN layers. The scattering network evaluation takes approximately five seconds at $L = 256$ (based on an unoptimized, single-threaded implementation). We do not observe any significant differences in speed of convergence between settings with and without the scattering network.

## 5.5 GAUSSIANITY OF THE COSMIC MICROWAVE BACKGROUND

In the standard inflationary cosmological model the temperature anisotropies of the cosmic microwave background (CMB), the relic radiation of the Big Bang, are predicted to be a realization of a Gaussian random field on the sphere. However, many alternative cosmological models predict weak deviations from Gaussianity. The statistical properties of the CMB are thus a powerful probe for distinguishing cosmological models and their study in observational data is of topical interest (e.g. Planck Collaboration VII, 2020; Planck Collaboration XXIII, 2014). We generate Gaussian and non-Gaussian simulations of the CMB (Rocha et al. 2005) at reso-

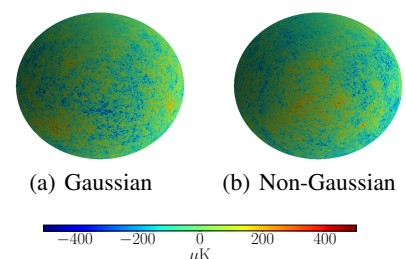

(a) Gaussian    (b) Non-Gaussian

−400   −200   0   200   400
$\mu$K

Figure 5: CMB simulations to be classified. For the weakly non-Gaussian simulations considered the classification problem is difficult; e.g. it is *not* possible to distinguish the simulations by eye.

lution $L = 1024$ (see Fig. 5). We train similar models to those considered for the MNIST experiment above to distinguish between CMB simulations (see Appendix B.4). Since the non-Gaussianity considered is weak this is a challenging classification problem. The first model, without a scattering network, achieves a classification accuracy of 53.1%, demonstrating the challenging nature of the problem. The second model, including a scattering network, achieves a classification accuracy of 95.3%. Since the scattering network provides a highly effective representational space, which captures high-frequency signal content in the low-resolution scattering coefficients, the model is able to achieve a considerably greater accuracy. The scattering network evaluation takes a little under two minutes at $L = 1024$ (based on an unoptimized, single-threaded implementation); recall that this need only be computed once, outside of training.

## 6 CONCLUSIONS

We have developed scattering networks on the sphere that yield a powerful and scalable representational space for spherical signals, providing an efficient signal representation that is rotationally equivariant, invariant to isometries up to a particular scale, stable to diffeomorphisms, and sensitive to all signal content, including high and low frequencies. When incorporated as an additional initial layer in the generalized spherical CNN framework, scattering networks allow spherical CNNs to be scaled to high-resolution data, while preserving rotational equivariance. In future work spherical scattering networks can be straightforwardly extended to spin signals and directional wavelets (by adopting spin scale-discretized spherical wavelets; McEwen et al. 2015c).

While Euclidean scattering networks have been shown to be highly effective and have been used in a number of studies, they nevertheless have not experienced pervasive use since Euclidean CNNs are already computational scalable. In the spherical setting, on the other hand, existing spherical CNNs are not computationally scalable. We therefore anticipate scattering networks, integrated as a layer in the generalized spherical CNN framework, to be of critical use to scale spherical CNNs to the high resolution data typical of many applications.

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

# A WAVELETS ON THE SPHERE

## A.1 REVIEW OF APPROACHES

Numerous wavelet frameworks on the sphere have been constructed. Early constructions were based either on fully continuous frameworks (e.g. Antoine & Vandergheynst, 1999; 1998; McEwen et al., 2006) or on fully discrete frameworks (Schröder & Sweldens, 1995; Barreiro et al., 2000; McEwen & Scaife, 2008; McEwen et al., 2011) based on the lifting scheme (Sweldens, 1997). While the former continuous frameworks lacked the ability to perfectly synthesize a signal from its wavelet coefficients in practice, the latter discrete frameworks lacked stable representations.

More recently, a number of exact discrete wavelet frameworks on the sphere have been developed with underlying continuous representations facilitated through the adoption of sampling theorems (e.g. McEwen & Wiaux, 2011a; McEwen et al., 2015a), including needlets (Narcowich et al., 2006; Baldi et al., 2009; Marinucci et al., 2008), directional scale-discretized wavelets (Wiaux et al., 2008; Leistedt et al., 2013; McEwen et al., 2015c; 2018) and the isotropic undecimated and pyramidal wavelet transforms (Starck et al., 2006). These approaches have been extended to spin functions on the sphere (Geller et al., 2008; Geller & Marinucci, 2010; 2011; Geller et al., 2009; McEwen et al., 2015c; 2014; Starck et al., 2009), extended to functions defined on the three-dimensional ball (Durastanti et al., 2014; Leistedt & McEwen, 2012; McEwen & Leistedt, 2013; Lanusse et al., 2012), and in some cases exhibit algorithms for efficient and exact computation (McEwen et al., 2007; 2013; 2015c).

Scale-discretized wavelet on the sphere (e.g. McEwen et al., 2018) have found widespread use in a variety of fields, including cosmology (e.g. Price et al., 2020; Rogers et al., 2016b;a; Leistedt et al., 2017; 2015) and geophysics (e.g. Marignier et al., 2021), and are well-suited for the construction of spherical scattering networks, as we elaborate in Appendix A.6. We therefore provide a concise review of the spherical scale-discretized wavelet framework and its properties.

## A.2 WAVELET ANALYSIS

While the scale-discretized wavelet transform on the sphere is reviewed very concisely in Section 2.2, we provide further details here. Since we provide a more complete review of the forward wavelet transform here, this subsection overlaps with the material of Section 2.2. However, proceeding subsections provide details not discussed in the main body of the article.

The scale-discretized wavelet transform of a function $f \in \mathrm{L}^2(\mathbb{S}^2)$ on the sphere $\mathbb{S}^2$ is defined by the convolution of $f$ with the *wavelet* $\psi_j \in \mathrm{L}^2(\mathbb{S}^2)$. The *wavelet coefficients* $W_j \in \mathrm{L}^2(\mathbb{S}^2)$ thus read

$$W_j(\omega) = (f \star \psi_j)(\omega) = \langle f, R_\omega \psi_j \rangle = \int_{\mathbb{S}^2} \mathrm{d}\mu(\omega') f(\omega')(R_\omega \psi_j)^*(\omega'), \tag{6}$$

where $\mathrm{d}\mu(\omega) = \sin\theta \mathrm{d}\theta \mathrm{d}\varphi$ denotes the rotationally invariant Haar measure on the sphere and $^*$ denotes complex conjugation. The wavelet scale $j$ encodes the angular localization of $\psi_j$, with increasing $j$ corresponding to smaller scale (i.e. higher frequency) signal content. The minimum and maximum wavelet scales considered are denoted $J_0$ and $J$ respectively, with $0 \leq J_0 \leq j \leq J$.

In general directional wavelets are supported by the scale-discretized wavelet framework (i.e. wavelets that are not azimuthally symmetric), resulting in wavelet coefficients that live on the rotation group $\mathrm{SO}(3)$; nevertheless, in this article we restrict our attention to axisymmetric wavelets (i.e. wavelets that are azimuthally symmetric). The wavelet coefficients capture high-frequency or bandpass information of the underlying signal; they do not capture low-frequency signal content.

A *scaling function* is introduced to capture low-frequency signal content. *Scaling coefficients* $\psi \in \mathrm{L}^2(\mathbb{S}^2)$ are given by convolution of $f$ with the scaling function $\phi \in \mathrm{L}^2(\mathbb{S}^2)$:

$$W(\omega) = (f \star \phi)(\omega) = \langle f, R_\omega \phi \rangle = \int_{\mathbb{S}^2} \mathrm{d}\mu(\omega') f(\omega')(R_\omega \phi)^*(\omega'). \tag{7}$$

## A.3 WAVELET SYNTHESIS

The signal $f$ can be synthesized perfectly from its wavelet and scaling coefficients by

$$f(\omega) = \int_{\mathbb{S}^2} \mathrm{d}\mu(\omega')W(\omega')(R_{\omega'}\phi)(\omega) + \sum_{j=J_0}^{J} \int_{\mathbb{S}^2} \mathrm{d}\mu(\omega')W_j(\omega')(R_{\omega'}\psi_j)(\omega), \tag{8}$$

provided that the following admissibility condition holds:

$$\frac{4\pi}{2\ell+1}\left[|\hat{\phi}_{\ell 0}|^2 + \sum_{j=J_0}^{J}|(\hat{\psi}_j)_{\ell 0}|^2\right] = 1, \quad \forall \ell, \tag{9}$$

where the spherical harmonic coefficients of the wavelet and scaling function are given by, respectively, $(\hat{\psi}_j)_{\ell m}\delta_{m0} = \langle \psi_j, Y_{\ell m}\rangle$ and $\hat{\phi}_{\ell m}\delta_{m0} = \langle \phi, Y_{\ell m}\rangle$.

## A.4 WAVELET CONSTRUCTION

Scale-discretized wavelets are constructed to satisfy the admissibility property of Equation 9 in order to ensure perfect signal synthesis and to be infinitely differentiable in harmonic space, resulting in excellent spatial localization properties.

Consider the infinitely differentiable Schwartz function with compact support $t \in [\lambda^{-1}, 1]$, for dilation parameter $\lambda \in \mathbb{R}_*^+, \lambda > 1$:

$$s_\lambda(t) \equiv s\left(\frac{2\lambda}{\lambda-1}(t-\lambda^{-1}) - 1\right), \tag{10}$$

where

$$s(t) \equiv \begin{cases} \exp\bigl(-(1-t^2)^{-1}\bigr), & t \in (-1, 1) \\ 0, & t \notin (-1, 1) \end{cases}. \tag{11}$$

Define the smoothly decreasing function $k_\lambda$ by

$$k_\lambda(t) \equiv \frac{\int_t^1 \frac{\mathrm{d}t'}{t'} s_\lambda^2(t')}{\int_{\lambda^{-1}}^1 \frac{\mathrm{d}t'}{t'} s_\lambda^2(t')}, \tag{12}$$

which is unity for $t < \lambda^{-1}$, zero for $t > 1$, and is smoothly decreasing from unity to zero for $t \in [\lambda^{-1}, 1]$. Define the wavelet kernel generating function by

$$\kappa_\lambda(t) \equiv \sqrt{k_\lambda(\lambda^{-1}t) - k_\lambda(t)}, \tag{13}$$

which has compact support $t \in [\lambda^{-1}, \lambda]$ and reaches a peak of unity at $t = 1$. The scale-discretized wavelet kernel for scale $j$ is then defined by

$$(\hat{\psi}_j)_{\ell m} = \sqrt{\frac{2\ell+1}{4\pi}} \kappa_\lambda(\lambda^{-j}\ell)\delta_{m0}, \tag{14}$$

which has compact support on $\ell \in [\lambda^{j-1}, \lambda^{j+1}]$ and reaches a peak of unity at $\lambda^j$. Scaling functions are required to probe the low-frequency content of the signal of interest not probed by the wavelets and are thus defined by

$$\hat{\phi}_{\ell m} = \sqrt{\frac{2\ell+1}{4\pi}} \sqrt{k_\lambda(\lambda^{-J_0}\ell)} \,\delta_{m0}. \tag{15}$$

The scaling function has compact support on $\ell \in [0, \lambda^{J_0}]$ and is unity up to $\lambda^{J_0-1}$, decaying smoothly to zero on $[\lambda^{J_0-1}, \lambda^{J_0}]$. An illustration of the harmonic support of the scaling function and wavelets is shown in Fig. 6.

While the dilation parameter $\lambda$ can be selected arbitrarily depending on the harmonic scales one wishes each wavelet to probe, provided $\lambda > 1$, a common choice is dyadic scaling with $\lambda = 2$. The maximum wavelet scale $J$ is set to ensure the wavelets reach the bandlimit $L$ of the signal of interest, yielding $J = \lceil \log_\lambda L \rceil$, where $\lceil \cdot \rceil$ is the ceiling function. The minimum wavelet scale $J_0$ may be freely chosen, provided $0 \le J_0 < J$. Alternatively, the minimum scale considered may be set by specifying a desired bandlimit $L_0$ for the scaling coefficients, yielding $J_0 = \lceil \log_\lambda L_0 \rceil$. For $J_0 = 0$ the wavelets probe the entire frequency content of the signal of interest except its mean, which is encoded in the scaling coefficients. The wavelet transform of a signal bandlimited at $L$ is therefore specified by the parameters $(L, \lambda, J_0)$, or alternatively $(L, \lambda, L_0)$.

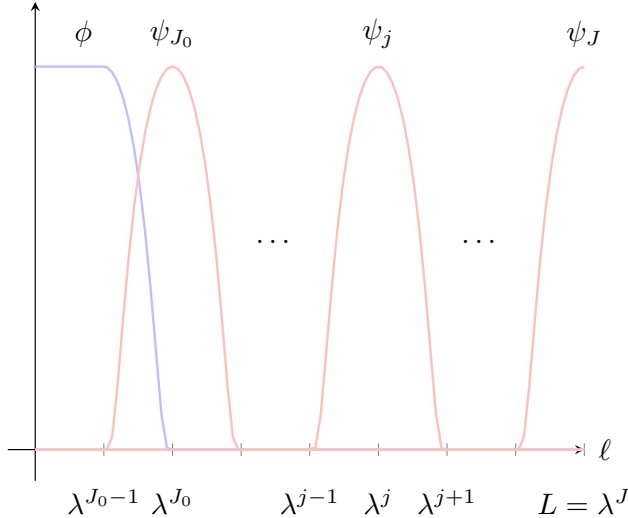

Figure 6: Harmonic support of scaling function $\phi$ and wavelets $\psi_j$ for scales $0 \leq J_0 \leq j \leq J$.

### A.5 Exact and Efficient Computation

The scale-discretized wavelet transform of signals on the sphere can be computed exactly and efficiently by appealing to sampling theorems on the sphere (e.g. McEwen & Wiaux, 2011b) (and rotation group for directional wavelets; McEwen et al. 2015b) and corresponding fast algorithms (McEwen et al., 2007; 2013; 2015c). By exploiting sampling theorems access to the underlying continuous signals is afforded and wavelet transforms can be computed in a manner that is theoretically exact, up to machine precision. We adopt the `s2let` [3] code implementing this scale-discretized wavelet transform on the sphere. For the axisymmetric wavelet transforms considered herein, the complexity of the wavelet transform scales as $\mathcal{O}(L^3)$. Further computational savings can be achieved by considering a *multi-resolution* setting where wavelet coefficients for scale $j$ are computed at the minimum resolution required to capture all information content, i.e. at $L = \lambda^{j+1}$ (Leistedt et al., 2013). We adopt the multi-resolution setting in our experiments since we found this does not markedly impact model equivariance or accuracy, while considerably reducing the computational time of wavelet transforms.

### A.6 Properties

As discussed, the computation of the scale-discretized wavelet transform is scalable. The wavelet transform is rotationally equivariant in theory, since it is based on spherical convolutions that themselves are clearly rotationally equivariance (shown in e.g. Cohen et al. 2018a), and in practice, since its computation leverages underlying sampling theory on the sphere. Moreover, scale-discretized spherical wavelets have excellent localization and stability properties, and constitute a Parseval frame (McEwen et al., 2018; 2015c). Finally, they are clearly sensitive to both low and high frequency signal content, with the scaling coefficients capturing low frequency content and wavelet coefficients capturing frequencies associated with the scale of the corresponding wavelet. Scale-discretized wavelets on the sphere thus provide an ideal wavelet framework on which to build spherical scattering networks.

## B Additional Information on Experiments

### B.1 Equivariance Tests

To test the equivariance of the scattering transform we perform similar experiments to those of Cobb et al. (2021, Appendix D), considering $N_f = 100$ random spherical signals $\{f_i\}_{i=1}^{N_f}$ with harmonic

---

[3]www.s2let.org

coefficients sampled from the standard normal distribution and $N_\rho = 100$ random rotations $\{\rho_j\}_{j=1}^{N_\rho}$ sampled uniformly on SO(3). To measure the extent to which an operator $\mathcal{A} : L^2(\mathbb{S}^2) \to L^2(\mathbb{S}^2)$ is equivariant we evaluate the mean relative error

$$\epsilon(\mathcal{A}(\mathcal{R}_{\rho_j} f_i), \mathcal{R}_{\rho_j}(\mathcal{A} f_i)) = \frac{1}{N_f} \frac{1}{N_\rho} \sum_{i=1}^{N_f} \sum_{j=1}^{N_\rho} \frac{\|\mathcal{A}(\mathcal{R}_{\rho_j} f_i) - \mathcal{R}_{\rho_j}(\mathcal{A} f_i))\|}{\|\mathcal{A}(\mathcal{R}_{\rho_j} f_i)\|} \tag{16}$$

resulting from pre-rotation of the signal, followed by application of $\mathcal{A}$, as opposed to post-rotation after application of $\mathcal{A}$, where the operator norm $\| \cdot \|$ is defined using the inner product $\langle \cdot, \cdot \rangle_{L^2(\mathbb{S}^2)}$.

## B.2 DIFFEOMORPHISM TESTS

To test the stability of spherical scattering networks to diffeomorphisms we compute differences in scattering network representations for spherical MNIST digits, averaged over 10,00 randomly selected digits. We define the difference between scattering network representations for spherical signals $f_i^{(k)}$, for class $k$ and instance $i$, by

$$\delta(\mathcal{S}_{\mathbb{P}_D} f_i^{(k_1)} - \mathcal{S}_{\mathbb{P}_D} f_j^{(k_2)}) = \frac{1}{N} \sum_{i,j}^{N} \|\mathcal{S}_{\mathbb{P}_D} f_i^{(k_1)} - \mathcal{S}_{\mathbb{P}_D} f_j^{(k_2)}\| , \tag{17}$$

where the norm $\| \cdot \|$ in this case denotes the inner product $\langle \cdot, \cdot \rangle_{L^2(\mathbb{S}^2)}$ applied to each scattering coefficient channel, then summed over all channels. For diffeomorphism tests we consider a spherical scattering network with parameters $(L, \lambda, L_0) = (128, 2, 32)$, descending paths only, and depth $D = 2$ (matching the settings of Section 5.4).

## B.3 SPHERICAL MNIST AT VARYING RESOLUTION

The first model considered for our MNIST classification experiments is the hybrid model proposed in Cobb et al. (2021). The first layer includes a directional convolution on the sphere that lifts the spherical input onto the rotation group. The second layer includes a convolution on the rotation group, followed by three layers comprising constrained generalized convolutions and tensor-product activations. The final restricted generalized convolution maps down to a rotationally invariant representation. As is traditional in convolution networks we gradually decrease the resolution, with $(L_0, L_1, L_2, L_3, L_4, L_5) = (20, 10, 10, 6, 3, 1)$, and increase the number of channels, with $(K_0, K_1, K_2, K_3, K_4, K_5) = (1, 20, 22, 24, 26, 28)$. We proceed these convolutional layers with a single dense layer of size 30, sandwiched between two dropout layers (keep probability 0.5), and then fully connect to the output of size 10. This model considers harmonic content of the spherical MNIST images up to $L = 32$ only (since it is too computationally demanding to run at high resolution). The resulting model has 59,396 learnable parameters.

The second model considered prepends a spherical scattering network to the first model. We consider a scattering network with $\lambda = 2$, descending paths only, and depth $D = 2$. The scattering network takes the spherical image at its original resolution and outputs scattering coefficients (channels) at resolution $L_0 = 32$. The non-scattering part of the model thus runs at the same resolution for both settings ($L = 32$). The resulting model has 60,396 learnable parameters for $L = 64$.

We train the network for 20 epochs on batches of size 32, using the Adam optimizer (Kingma & Ba, 2015) with a decaying learning rate starting at 0.001. For the restricted generalized convolutions we follow the approach of Kondor et al. (2018) by using $L_1$ regularization (regularization strength $10^{-5}$) and applying a restricted batch normalization across fragments, where the fragments are only scaled by their average and not translated (to preserve equivariance).

We adopt the same core hybrid model, with the same parameters, for both scenarios with and without the scattering network. The only different is the number of input channels since for the case without scattering we have one greyscale spherical image of a digit, whereas for the scattering case we have a channel for each set of scattering coefficients (i.e. each blue node in Fig. 1). This difference in number of input channels results in a small difference in overall number of learnable parameters for the two models, since a different convolutional filter is applied to each of the input channels. To ensure this does not have a significant impact on results, we also consider a hybrid model without a

scattering network with $K_1 = 21$ (instead of $K_1 = 20$) channels in the first layer, which results in a model with 60,507 learnable parameters (more learnable parameters than the model with a scattering network for $L = 64$). We find no significant difference between the models with $K_1 = 20$ and $K_1 = 21$, suggesting the superiority of the approach with a scattering network in Table 2 is due to the effectiveness of the scattering network in mixing high-frequency signal content to low frequencies, rather than the very small increase in number of parameters of the models that include a scattering network.

### B.4 GAUSSIANITY OF THE COSMIC MICROWAVE BACKGROUND

To simulate CMB observations we adopt the Lambda Cold Dark Matter (LCDM) CMB power spectrum that best fits CMB observations made by the ESA Planck satellite (Planck Collaboration I, 2020). CMB simulations are generated following the approach of Rocha et al. (2005), where the non-Gaussian distribution is derived from the wavefunctions of the harmonic oscillator. We set $\alpha_3 = 0.0$ to recover Gaussian simulations and $\alpha_3 = 0.2$ to recover non-Gaussian simulations(following Rocha et al. 2005).

We consider simulations at resolution $L = 1024$, training on 1024 simulations and evaluating on a test set of 128 simulations. We consider similar models and training configuration as the MNIST experiment presented above but with scattering paths such that $j_2 = j_1 - 1$ (to avoid generating a large number of output channels and to promote mixing from high to low frequencies) and train for 200 epochs.

