# OpenReview forum: "Scattering Networks on the Sphere for Scalable and Rotationally Equivariant Spherical CNNs"
_ICLR.cc/2022/Conference — ICLR 2022 Poster_

### Official Review · Reviewer_FtPm · 2021-11-02

**Correctness:** 4
**Technical Novelty And Significance:** 3
**Empirical Novelty And Significance:** 3
**Recommendation:** 6
**Confidence:** 4

**Main Review:**

*Strengths*

The paper is very well written. The contributions are clearly and concisely explained, the approach is well motivated and the relation with existing work is well described. The paper proposes an elegant solution to the practical problem of scaling spherical CNNs to large resolution inputs.

*Weaknesses*

1) The scattering transform utilized has complexity $\mathcal{O}(L^3)$ and consists of cascaded convolutions with axisymmetric wavelets and nonlinearities. The spherical convolution as defined by Driscoll and Healy [2] also considers axisymmetric (zonal) filters and can also be computed in $\mathcal{O}(L^3)$ while faster methods scaling as $\mathcal{O}(L^2 (\log L)^{2})$ for the forward transform also exist. The spherical CNNs of Esteves et al [3] is based on such faster spherical convolutions with zonal filters. What would be the effect of using this faster kind of axisymmetric spherical CNNs instead of the scattering transform? It seems to me that the learnable instead of fixed filters might be advantageous, but they might also be more expensive because pre-computing would not possible. In this case, perhaps using cascaded Driscoll-Healy convolutions with random, localized axisymmetric filters might make sense? Please elaborate on such comparisons; it would be even better to provide quantitative results in terms of accuracy and training/inference speed.

2) For the experiments on rotated MNIST (section 5.3), what is the number $n$ of channels after the scattering transform? Since the baseline starts from a single channel at $L=32$, I believe there are two factors that explain the better performance of the scattering pre-processing: 1) the preservation of high-frequency content and 2) the larger number of filters at the first layer. I am assuming that the first non-scattering layer has the same number of channels $c$ for both models, which would translate into $c$ filters for the non-scattering model but $cn$ filters for the scattering. I suggest an extra baseline to disentangle the two effects: use $cn$ channels followed by a nonlinearity and projection to $c$ channels on the first layer for the non-scattering model. It would also be interesting to see the effects of the scattering pre-processing for $L=32$ -- I wonder if it can improve performance even on lower resolutions, which would make it more widely applicable.

3) There is no mention of a code release in the paper. It seems that the generalized spherical CNN of Cobb et al [1] is used in the experiments, for which there is no public codebase available as far as I know. So releasing the code of this submission, besides ensuring reproducibility of the results claimed, would also provide the community with an easy way to build upon [1].

*References*

[1] Cobb et al, "Efficient Generalized Spherical CNNs", ICLR'21.

[2] Driscoll and Healy. "Computing Fourier Transforms and Convolutions on the 2-Sphere" (1994).

[3] Esteves et al, "Learning SO(3) Equivariant Representations with Spherical CNNs", ECCV'18.


**Summary Of The Paper:**

The paper proposes a method to scale spherical CNNs to high resolution inputs. This is an important problem since the high computational cost prevents the use of spherical CNNs in many applications. The solution proposed is to define a spherical scattering network to pre-process the high resolution inputs, producing a lower-resolution channels that still carry some of the high frequency information. This is shown to improve performance when compared with a simple downsampling of the high resolution inputs.


**Summary Of The Review:**

This is a good, well written paper that proposes an elegant and principled way of scaling spherical CNNs to high resolution inputs. My suggestions are mostly in the direction of making the experimental section more convincing and reproducible, and I will be happy to increase my score in case they are addressed.

---

> ### Author Response · Authors · 2021-11-22
> **Response to Reviewer FtPm (1 of 2)**
>
> We thank the Reviewer for their comments.  We respond to each comment in turn below.  The Reviewer's original comments are italicized, while our responses are given in Roman font.  All revisions to the manuscript are highlighted in red.
>
> *The scattering transform utilized has complexity $\mathcal{O}(L^3)$ and consists of cascaded convolutions with axisymmetric wavelets and nonlinearities. The spherical convolution as defined by Driscoll and Healy [2] also considers axisymmetric (zonal) filters and can also be computed in $\mathcal{O}(L^3)$ while faster methods scaling as $\mathcal{O}(L^2 (\log L)^2)$  for the forward transform also exist. The spherical CNNs of Esteves et al [3] is based on such faster spherical convolutions with zonal filters. What would be the effect of using this faster kind of axisymmetric spherical CNNs instead of the scattering transform? It seems to me that the learnable instead of fixed filters might be advantageous, but they might also be more expensive because pre-computing would not possible. In this case, perhaps using cascaded Driscoll-Healy convolutions with random, localized axisymmetric filters might make sense? Please elaborate on such comparisons; it would be even better to provide quantitative results in terms of accuracy and training/inference speed.*
>
> The dominant component of the computation for both axisymmetric convolutions and wavelet transforms is the spherical harmonic transform.  While it is indeed the case that algorithms to compute the spherical harmonic transform have been developed by [Driscoll & Healy (1999)](https://www.sciencedirect.com/science/article/pii/S0196885884710086) that scale as $\mathcal{O}(L^2 (\log L)^2)$, it is known that such algorithms suffer from stability problems ([Healy et al. 2003](https://link.springer.com/article/10.1007/s00041-003-0018-9), [Kostelec & Rochmore 2008](https://link.springer.com/article/10.1007/s00041-008-9013-5)).  In [McEwen & Wiaux (2011)](https://arxiv.org/abs/1110.6298) comparisons of the accuracy (Fig. 4) and computation time (Fig. 5) of alternative spherical harmonic transforms are shown, where the methods of Driscoll, Healy and collaborations are demonstrated to fail above bandlimit $L=1024$.
>
> While [Esteves et al. (2018)](https://arxiv.org/abs/1711.06721) demonstrate excellent performance with spherical CNNs based on axisymmetric convolutions, the resolution of the problems considered is relatively low and hence the faster $\mathcal{O}(L^2 (\log L)^2)$ spherical convolutions are appropriate. If these models were to be scaled to higher resolutions, stable $\mathcal{O}(L^3)$ spherical harmonic transforms would need to be adopted (e.g. either those of [Driscoll & Healy 1999](https://www.sciencedirect.com/science/article/pii/S0196885884710086) or [McEwen & Wiaux 2011](https://arxiv.org/abs/1110.6298)).
>
> In any case, if more efficient spherical harmonic transforms that were stable to high resolution were developed, they could be adopted for the axisymmetric scattering transform on the sphere presented in the current manuscript.  In principle, therefore, there is no real difference between the complexity of the layers developed by [Esteves et al. (2018)](https://arxiv.org/abs/1711.06721) and presented in this work: if they were to adopt the same spherical harmonic transforms, then complexity would be identical.
>
> However, spherical harmonic transforms at high resolution are computationally demanding in practice (at either $\mathcal{O}(L^2 (\log L)^2)$ or $\mathcal{O}(L^3)$).  The spherical scattering approach presented in the current manuscript has the additional advantage that it does not include parameters that need to be learnt and so can be performed as a pre-processing step.  Moreover, the modulus function that appears in the scattering networks is better at preserving rotational equivariance than the ReLU activation functions typically used in spherical CNNs (see Section 5.2 of the manuscript).

---

> ### Author Response · Authors · 2021-11-22
> **Response to Reviewer FtPm (2 of 2)**
>
> *For the experiments on rotated MNIST (section 5.3), what is the number  of channels after the scattering transform? Since the baseline starts from a single channel at $L=32$, I believe there are two factors that explain the better performance of the scattering pre-processing: 1) the preservation of high-frequency content and 2) the larger number of filters at the first layer. I am assuming that the first non-scattering layer has the same number of channels $c$ for both models, which would translate into $c$ filters for the non-scattering model but $cn$ filters for the scattering. I suggest an extra baseline to disentangle the two effects: use  channels followed by a nonlinearity and projection to  channels on the first layer for the non-scattering model. It would also be interesting to see the effects of the scattering pre-processing for $L=32$ -- I wonder if it can improve performance even on lower resolutions, which would make it more widely applicable.*
>
> The Reviewer makes a very good point here.  As the Reviewer suggests, there is a small difference in number of parameters between the models with and without a scattering network.  The difference arises from the differing number of input channels for the case with the scattering network, where a different convolutional filter is then applied to each of the input channels.  The model without the scattering network has 59,396 learnable parameters, whereas the model with the scattering network has 60,396 learnable parameters.  Since the difference in number of parameters is small ($\sim 1.7$%) we do not expect this to have a significant impact on classification accuracy but it is nevertheless important to check.  We now consider a third model, modifying the model without the scattering network to have more channels in the first layer so that its number of learned parameters is greater than the model with the scattering network.  This involves changing the number of channels in the first layer from $K_1=20$ to $K_1=21$, which results in a model with 60,507 learnable parameters.  Indeed, we do not observe any significant improvement in classification accuracy with this change.  This additional experiment is discussed in Appendix B.3.  Moreover, additional details about the model have been added so that its full architecture is explicit, which also helps in describing the change to number of channels made in the third model.
>
> *There is no mention of a code release in the paper. It seems that the generalized spherical CNN of Cobb et al [1] is used in the experiments, for which there is no public codebase available as far as I know. So releasing the code of this submission, besides ensuring reproducibility of the results claimed, would also provide the community with an easy way to build upon [1].*
>
> We completely agree with the Reviewer and intend to make the code public as soon as possible (as we have done for many other codes).  There are commercial constraints stopping us doing so at present but we hope these to be removed soon, in which case we will indeed make the code public.  In the interests of reproducibility in the interim, we have also added a number of further details about the experiments in the new Appendix B added to the manuscript.

---

> ### Comment · Reviewer_FtPm · 2021-11-27
> **Response to rebuttal.**
>
> Thank you for the response and explanations about the stability and speed trade-offs. I maintain my recommendation for acceptance.

---

### Official Review · Reviewer_Zc9b · 2021-11-02

**Correctness:** 3
**Technical Novelty And Significance:** 3
**Empirical Novelty And Significance:** 3
**Recommendation:** 6
**Confidence:** 3

**Main Review:**

##########################################################################

Summary:
The paper proposes an approach to scale rotationally equivariant convolutions on spherical domains to work with signals with arbitrary resolutions. This is accomplished by leveraging the properties of the scattering transform, by converting the input signal to the corresponding wavelet-based representation, before feeding it to some arbitrary spherical CNN architecture. This can be considered as a preprocessing step for the data, as the transform does not contain any learnable parameters. The obtained representations are isometry invariant up to a given scale and stable to small diffeomorphisms proved theoretically extending the proof in [Perlmutter et al. (2020)]. The approach is validated experimentally by testing the properties of rotation equivariance, as well as being agnostic to resolution on synthetic datasets, and the frequency coverage of the representation. enabling to scale up to high resolutions for the input signals thanks to the compactness of the representation and showing a significant improvement over using a harmonic representation.


##########################################################################

Reasons for score:



Overall, I am leaning toward acceptance of the paper.  I like the simplicity of the idea and its effectiveness. Nevertheless, I find that some conceptual points in the cons section need to be addressed, in order to make the paper clearer in some of its parts. Hopefully, the authors can address my concerns in the rebuttal period.

##########################################################################
Pros:


1. The paper improves current approaches to apply convolution on spherical domains to high resolutions by leveraging the compactness of the scattering transform representation.

2. The idea is simple and yet provides several advantages over existing approaches, as shown in the experimental section.

 3. The paper is well written and enough references and context are provided for non-experts in the field.




##########################################################################

Cons and questions:


1. It is not entirely clear to me how the CNNs can operate in a rotation equivariant way if the representation obtained from the scattering transform is invariant (up to a scale) to isometries since the latter class of transformations contains the former. Wouldn't in this case the CNN be invariant to rotations as well? Please clarify this point.

2. It would be interesting to show statistics about time for processing the scattering transform and training the network on top of this representation: E.g. does the network converge faster applying the scattering transform?

3. Please, to improve the self-containment of the manuscript, report somewhere (could be in the Appendix) the definition of the equivariance error metric as well as the details for the equivariance test.


4. Concerning the spherical MNIST experiment:

    (1) It would be interesting to report results also on the test set with no rotated digits, to better evaluate only the amount of invariance to changes in resolutions.

    (2) Does the network applied to harmonic representations have the same number of parameters in the first layer (I.e. the number of input channels is the same) with respect to the one applied to the scattering transform?

    (3)  It would be interesting as well  to test the robustness to diffeomorphisms experimentally on  this dataset


##########################################################################

Questions during rebuttal period:


Please address and clarify the cons and questions above



#########################################################################

I spotted some typos:

(1) Page 2 section 1:  are there advantages-> there are advantages

(2) section 2.2: except its mean-> except for its mean

(3) section 2.3: elaborate these -> elaborate on these

(4) section 2.3: a isometric -> an isometric

(5) section 3: which follow by direct -> which follows by a direct

(6) Figure 1, caption: propagated though -> propagated through

(7) section 5.2: after application -> after the application

(8) section 5.3: becomes increasing -> becomes increasingly

(9) section 5.3: in following spherical -> in the following spherical





**Summary Of The Paper:**

The paper proposes an approach to scale rotationally equivariant convolutions on spherical domains to work with signals with arbitrary resolutions. This is accomplished by leveraging the properties of the scattering transform, by converting the input signal to the corresponding wavelet-based representation, before feeding it to some arbitrary spherical CNN architecture. This can be considered as a preprocessing step for the data, as the transform does not contain any learnable parameters. The obtained representations are isometry invariant up to a given scale and stable to small diffeomorphisms proved theoretically extending the proof in [Perlmutter et al. (2020)]. The approach is validated experimentally by testing the properties of rotation equivariance, as well as being agnostic to resolution on synthetic datasets, and the frequency coverage of the representation. enabling to scale up to high resolutions for the input signals thanks to the compactness of the representation and showing a significant improvement over using a harmonic representation.


**Summary Of The Review:**

Overall, I am leaning to accept the paper. I like the simplicity of the idea and its effectiveness. Nevertheless, I find that some of the points in the cons section, need to be addressed, in order to make the paper clearer in some of its parts. Hopefully, the authors can address my concern in the rebuttal period.

---

> ### Author Response · Authors · 2021-11-22
> **Response to Reviewer Zc9b (1 of 2)**
>
> We thank the Reviewer for their comments.  We respond to each comment in turn below.  The Reviewer's original comments are italicized, while our responses are given in Roman font.  All revisions to the manuscript are highlighted in red.
>
> *It is not entirely clear to me how the CNNs can operate in a rotation equivariant way if the representation obtained from the scattering transform is invariant (up to a scale) to isometries since the latter class of transformations contains the former. Wouldn't in this case the CNN be invariant to rotations as well? Please clarify this point.*
>
> The Reviewer makes a very good point here, which we have clarified below and in the manuscript.  The key subtlety here is invariance ***up to a scale***.  In the spherical scattering network presented in the current manuscript, the scale of invariance is controlled by the bandlimit of the wavelet scaling function $L_0$, which is in turn controlled by the minimum wavelet scale $J_0$ considered (see Appendix A of manuscript).  In the case where $J_0=0$, corresponding to $L_0=1$, the wavelet scaling function is constant over the sphere and the scattering coefficients (which, recall, are given by the wavelet scaling coefficients at each layer) are simply the mean of the signal at the previous layer.  In this case, complete invariance to rotations is achieved, as the Reviewer comments.  However, for choices of $J_0>0$, i.e. $L_0>1$, the scaling coefficients are given by a local averaging that does not include the entire sphere.  Consequently, the resulting scattering coefficients are not completely invariant to rotations.  For large $J_0$ (large $L_0$) the scaling function has small support and only very localised invariance is introduced.  Rotational equivariance, on the other hand, is independent of the choice of the minimum wavelet scale $J_0$ and holds for all settings since the wavelet scattering transform is simply composed of cascades of spherical convolutions (wavelet transforms) or pointwise applications of the absolute value function, both of which are themselves rotationally equivariant.  Formally, in the $J_0=0$ setting where the network is fully invariant to rotations, equivariance still holds, it is just that the spherical scattering coefficients for each path are reduced to a single constant value over the sphere.  In practice we ensure $J_0\neq0$ so that the scattering network is not completely invariant to rotations, before signals are passed to the following spherical CNN layers.
>
> *It would be interesting to show statistics about time for processing the scattering transform and training the network on top of this representation: E.g. does the network converge faster applying the scattering transform?*
>
> Computational times were not previously given since application of the spherical scattering network is independent of training and is performed as a data-preprocessing step.  Execution and training of the subsequent hybrid spherical CNN that follows the scattering network is then the same as before when running at low resolution.  Nevertheless, while computation time of the scattering network does not impact training time, we agree readers may be interested in the the general time required to evaluate spherical scattering networks for the precompution.  This is approximately five seconds for the highest MNIST resolution considered ($L=256$) and a little under two minutes for the CMB resolution considered ($L=1024$).  However, we have *not* implemented a parallelized version of the spherical scattering transform: the current implementation runs sequentially on a single CPU core.  As mentioned, the scattering network evaluation is trivially parallelized in principle, hence there is great scope for reducing computation time.  For training of MNIST, the network takes roughly five minutes per epoch and so training is completed in a little under two hours for 20 epochs.  We do not notice any significant differences in speed of convergence between settings with and without the scattering network.  Related discussions have been added to the manuscript.
>
> *Please, to improve the self-containment of the manuscript, report somewhere (could be in the Appendix) the definition of the equivariance error metric as well as the details for the equivariance test.*
>
> Further details regarding the equivariance error metric, and generally regarding the numerical experiments, have been added in Appendix B.

---

> ### Author Response · Authors · 2021-11-22
> **Response to Reviewer Zc9b (2 of 2)**
>
> *Concerning the spherical MNIST experiment:*
>
> *(1) It would be interesting to report results also on the test set with no rotated digits, to better evaluate only the amount of invariance to changes in resolutions.*
>
> These experiments have been run and added to the results presented in Table 2.
>
> *(2) Does the network applied to harmonic representations have the same number of parameters in the first layer (I.e. the number of input channels is the same) with respect to the one applied to the scattering transform?*
>
> The Reviewer makes a very good point here.  As the Reviewer suggests, there is a small difference in number of parameters between the models with and without a scattering network.  The difference arises from the differing number of input channels for the case with the scattering network, where a different convolutional filter is then applied to each of the input channels.  The model without the scattering network has 59,396 learnable parameters, whereas the model with the scattering network has 60,396 learnable parameters.  Since the difference in number of parameters is small ($\sim 1.7$%) we do not expect this to have a significant impact on classification accuracy but it is nevertheless important to check.  We now consider a third model, modifying the model without the scattering network to have more channels in the first layer so that its number of learned parameters is greater than the model with the scattering network.  This involves changing the number of channels in the first layer from $K_1=20$ to $K_1=21$, which results in a model with 60,507 learnable parameters.  Indeed, we do not observe any significant improvement in classification accuracy with this change.  This additional experiment is discussed in Appendix B.3.  Moreover, additional details about the model have been added so that its full architecture is explicit, which also helps in describing the change to the number of channels made in the third model.
>
> *(3) It would be interesting as well to test the robustness to diffeomorphisms experimentally on this dataset*
>
> This is an interesting idea.  Having given this some further thought, we have devised the following experiment to test this empirically.  Taking the prototypical example of classification of hand-written digits, we compute differences in scattering network representations for both inter- and intra-class instances using MNIST digits projected onto the sphere. Small diffeomorphisms are likely responsible for intra-class variations and large diffeomorphisms for inter-class variations.  Furthermore, by Theorem 2, a larger diffeomorphism leads to a larger difference in scattering representation.  For MNIST digits, we therefore expect the average difference between scattering representations to increase as the size of diffeomorphisms within and between digits increases.  A new section (Section 5.3) has been added to the manuscript describing this experiment and presenting the results.  A new appendix (Appendix B.2) has also been added to the manuscript, giving further details on the experiment and the scattering representation difference metric computed.
>
> The results of this experiment match our intuitive expectation, e.g. we find intra-class scattering representation differences to be smaller than inter-class differences and digits that are morphologically similar appear to have smaller scattering representation differences than digits that are morphologically dissimilar, although it is difficult to quantify the size of a diffeomorphism mapping one digit to another.  These results are discussed further in Section 5.3 and support the theoretical result presented in the manuscipt that spherical scattering networks are stable to diffeomorphisms.
>
> This revision required adding a new section to the manuscript.  To fit this into the main body a number of minor modifications were made elsewhere and some material regarding the details of other experiments has been moved to appendices.  Furthermore, a number of further details about experiments have been added to the new appendices.
>
> *Typos*
>
> We thank the Reviewer very much for catching a number of typos, all of which have now been corrected.

---

> > ### Comment · Reviewer_Zc9b · 2021-11-29
> > **Response to authors rebuttal**
> >
> > I thank the authors for their response and their efforts during the rebuttal period.
> >
> > All my questions and issues have been addressed and clarified. I appreciated the additional experiments performed in the rebuttal period: as a minor side note, could you please specify that the scores in Figure 3 are relative to the smallest error in intraclass variations of the digit 1 (if this is indeed the case) and report also the corresponding real number for the latter?
> >
> > Overall, I believe that the paper brings advantages and insights to the field and for these reasons I confirm my initial intention to accept it.

---

> > > ### Author Response · Authors · 2021-11-30
> > > **Response to Reviewer response**
> > >
> > > In terms of the additional experiment and the results in Figure 3, the Reviewer's interpretation is indeed correct. The normalisation with respect to digit one is described in the main text but we agree it would be useful to also include this in the figure caption and will update the manuscript accordingly.  The raw number for digit one is 0.11713896, which we will also quote in the manuscript.

---

### Official Review · Reviewer_74vz · 2021-11-07

**Correctness:** 4
**Technical Novelty And Significance:** 2
**Empirical Novelty And Significance:** 1
**Recommendation:** 5
**Confidence:** 4

**Main Review:**

Strengths:
-	As the precomputation step can be done more efficiently than a some spherical CNNs, this step allows such methods to scale to higher-resolution data.

Weaknesses:
-	The authors claim that scattering networks yield rotationally invariant features. It seems natural to me to have a pre-computation step for a spherical CNN generate spherical or SO(3) signals. So how are the scattering network outputs used? Are they added as a constant signal to the sphere as extra channels? If so, why didn’t the authors explore pre-computation steps that result in non-constant signals? Please let me know if I misunderstand this.
-	The authors claim that some methods, like DeepSphere [Perraudin 2019] are not equivariant. Can the authors clarify their claim, as this directly contradicts the claims in that paper? As this graph-based method seem very scalable to high resolutions, I don’t see why DeepSphere doesn’t solve the problem the authors invent a new method for.
-	The experimental section is very weak. The authors cite many other spherical CNN papers, but only compare to a single instance of one other paper. This is insufficient in informing the reader when the proposed method is best used. For example, why didn’t the authors compare to the DeepSphere method, which also experiments on CMB data?
-	I find the presentation of the wavelets and scattering transform unclear. For example, it doesn’t say how explicitly the wavelets are constructed. Also, I don’t follow the discussion of the dilation parameter.
-	The method has limited novelty: doing a pre-computation step before applying a neural networks is very widely explored. Also, the theoretical contributions seem incremental changed to previous theoretical results.

Minor points:
-	The usage of $w$ an $\omega$ together is confusing.
-	Why is the output of the convolution with the wavelet a spherical signal, rather than a SO(3) signal? Does the wavelet contain a SO(2) symmetry?
-	In thm 1, $V_\zeta$ is undefined.


**Summary Of The Paper:**

The authors propose to process signals on the sphere via an equivariant CNN where, as a pre-processing step, a scattering network is used. As these scattering networks can be computed efficiently at high resolutions, this allows for spherical CNNs to be used on higher-resolution signals.
As experiments, the authors compare a spherical CNN with and without the scattering network pre-computation step on spherical MNIST and a cosmic background radiation dataset and find that the pre-computation step aids performance.


**Summary Of The Review:**

As far as I can tell, the authors do a pre-computation step which results in a constant signal over the sphere, which I find an odd choice.  The experimental section is severely lacking, and the method has limited novelty. Hence, I cannot recommend acceptance.

Updated my score to a 5.

---

> ### Author Response · Authors · 2021-11-22
> **Response to Reviewer 74vz (1 of 2)**
>
> We thank the Reviewer for their comments.  We respond to each comment in turn below.  The Reviewer's original comments are italicized, while our responses are given in Roman font.  All revisions to the manuscript are highlighted in red.
>
> *The authors claim that scattering networks yield rotationally invariant features. It seems natural to me to have a pre-computation step for a spherical CNN generate spherical or SO(3) signals. So how are the scattering network outputs used? Are they added as a constant signal to the sphere as extra channels? If so, why didn’t the authors explore pre-computation steps that result in non-constant signals? Please let me know if I misunderstand this.*
>
> This is indeed an important point and we have clarified the related discussion in the manuscript.
>
> The scattering network is not generally invariant to rotations.  The network is invariant up to a scale.  In the spherical scattering network presented in the current manuscript, the scale of invariance is controlled by the bandlimit of the wavelet scaling function $L_0$, which is in turn controlled by the minimum wavelet scale $J_0$ considered (see Appendix A of manuscript).
>
> For choices of $J_0>0$ ($L_0>1$), which is the case considered in practice, the scaling coefficients are given by a local averaging that does not include the entire sphere.  The resulting scattering coefficients are then signals defined over the sphere that are not constant, i.e. the network is not completely invariant to rotations.  The output of the scattering network (the scattering coefficients) is thus a collection of (non-constant) spherical signals for different paths.  We associate each path with a different channel and then feed the scattering coefficients into a hybrid spherical CNN.
>
> In principle the output of the scattering network can be made fully invariant to rotations by setting $J_0=0$, corresponding to $L_0=1$.  Then the wavelet scaling function is constant over the sphere and the scattering coefficients are simply the mean of the signal at the previous layer.  However, this is not likely to lead to an effective representation space on which to build further spherical CNN layers and so we do not adopt this configuration in numerical experiments.  Full invariance to rotations, if desired, is instead introduced in final layers of the hybrid spherical CNN.
>
> *The authors claim that some methods, like DeepSphere [Perraudin 2019] are not equivariant. Can the authors clarify their claim, as this directly contradicts the claims in that paper? As this graph-based method seem very scalable to high resolutions, I don’t see why DeepSphere doesn’t solve the problem the authors invent a new method for.*
>
> The DeepSphere approach of [Perraudin et al. (2019)](https://arxiv.org/abs/1810.12186) is not strictly equivariant.  Indeed in that article the authors state: "DeepSphere scales as $\mathcal{O}(N_\text{pix})$ at the expense of not being exactly equivariant" ([Perraudin et al. 2019](https://arxiv.org/abs/1810.12186)).  It is well-know that a completely regular point distribution on the sphere does in general not exist ([Tegmark 1996](https://arxiv.org/abs/astro-ph/9610094)).  Consequently, it is impossible for any approach based on a discrete representation of the sphere, such as graph neural networks, to achieve strict equivariance (see [Cobb et al. 2021](https://arxiv.org/abs/2010.11661) for further discussion).  It is well-known that when the graph approximates some underlying manifold, in the limiting case of increasing graph sample density (i.e. "resolution") the eigenfunctions of the graph Laplacian approach those of underlying manifold (e.g. [Singer 2006](https://www.sciencedirect.com/science/article/pii/S1063520306000510)). In this limiting case, strict equivariance is achieved due to the connection to the representation of the underlying continuous manifold.  However, in practice this limiting situation is clearly not realized and so equivariance is inexact.   While graph-based approaches such as DeepSphere are of great use, in particular they can be computed efficiently (as the Reviewer comments and as is discussed in the introduction of the manuscript), they do not achieve strict equivariance.  Furthermore, with such approaches the accuracy at which equivariance can be achieved is tightly coupled to resolution.  In contrast, the proposed spherical scattering network achieves strict equivariance for all resolutions.  In addition, DeepSphere is limited to isotropic filters, limiting representational capacity, which is not the case for the approach consider in the current manuscript.  Thus, graph-based approaches such as DeepSphere and the proposed spherical scattering networks are clearly complementary rather than alternatives approaches to solve the same problem.

---

> ### Author Response · Authors · 2021-11-22
> **Response to Reviewer 74vz (2 of 2)**
>
> *The experimental section is very weak. The authors cite many other spherical CNN papers, but only compare to a single instance of one other paper. This is insufficient in informing the reader when the proposed method is best used. For example, why didn’t the authors compare to the DeepSphere method, which also experiments on CMB data?*
>
> The purpose of the current manuscript is to develop an approach to scale spherical CNNs to high resolution data that is also rotationally equivariant.  As discussed in the response to the Reviewer's comment above, this can only be achieved through a connection to the underlying continuous representation of the sphere.  The only spherical CNN approaches that take this approach and thus achieve accurate rotational equivariance are those of [Cohen et al. 2018](https://arxiv.org/abs/1801.10130), [Esteves et al. 2018](https://arxiv.org/abs/1711.06721), [2020](https://arxiv.org/abs/1711.06721), [Kondor et al. 2018](https://arxiv.org/abs/1806.09231) and [Cobb et al. 2021](https://arxiv.org/abs/2010.11661).  Furthermore, the approach of [Cobb et al. 2021](https://arxiv.org/abs/2010.11661) is general and includes all of the others as special cases.  One can thus consider comparison with [Cobb et al. 2021](https://arxiv.org/abs/2010.11661) to capture all of these cases.  Indeed, the hybrid network considered in the current manuscript includes layers based on those of [Cohen et al. 2018](https://arxiv.org/abs/1801.10130), [Esteves et al. 2018](https://arxiv.org/abs/1711.06721), and [Kondor et al. 2018](https://arxiv.org/abs/1806.09231).
>
> It is indeed the case that we could have considered comparison with graph-based approaches such as DeepSphere and others, however as discussed we would not be comparing similar methodologies since rotational equvariance is inexact for graph-based approaches.  We therefore limited comparisons to alternative approaches that are rotationally equivariant.
>
> *I find the presentation of the wavelets and scattering transform unclear. For example, it doesn’t say how explicitly the wavelets are constructed. Also, I don’t follow the discussion of the dilation parameter.*
>
> Due to space constraints the discussion of the wavelet construction is presented in Appendix A, rather than in the main body of the manuscript.  These wavelets are presented in other published articles and so we decided against taking further space from the main body to review these works.  Rather, we refer the reader unfamiliar with wavelets on the sphere to these articles and present a concise summary and discussion in the context of spherical scattering networks in the appendix.  Other Reviewers have found the discussion of the manuscript clear, commenting "It is a nicely written paper with a thorough explanation of the theory behind scattering networks" and "The paper is very well written. The contributions are clearly and concisely explained, the approach is well motivated and the relation with existing work is well described."
>
> *The method has limited novelty: doing a pre-computation step before applying a neural networks is very widely explored. Also, the theoretical contributions seem incremental changed to previous theoretical results.*
>
> Contrary to the Reviewer's comment, there are no alternative spherical CNNs approaches that suggest a pre-computation to reach higher resolutions, while also exhibiting strict rotational equivariance.  We agree that the theoretical contributions of the manuscript, i.e. Theorem 1 and Theorem 2, follow directly from [Perlmutter et al. (2020)](https://arxiv.org/abs/1905.10448), as we clearly highlight.  The main novelty of the manuscript is the construction and use of scattering networks on the sphere to scale spherical CNNs to high-resolution settings, whilst preserving rotational equivariance and providing an effective representational space on which to build.  No alternative rotationally equivariant spherical CNN approach is able to scale to the resolutions that are made accessible by the techniques proposed in the manuscript.
>
> *The usage of $w$ an $\omega$ together is confusing.*
>
> We have replaced $w$ by $W$ to avoid this confusion.
>
> *Why is the output of the convolution with the wavelet a spherical signal, rather than a SO(3) signal? Does the wavelet contain a SO(2) symmetry?*
>
> The output of the convolution of a wavelet with a spherical signal, i.e. the wavelet coefficients, are defined on the sphere rather than SO(3) since we adopt axisymmetric wavelets that are azimuthally symmetric.  Spherical scattering networks can be straightforwardly extended to directional wavelets, resulting in wavelet coefficients on SO(3), as discussed in the conclusions of the manuscript.
>
> *In thm 1, $V_\eta$ is undefined.*
>
> $V_\eta$ is defined in Section 2.3

---

> ### Comment · Reviewer_74vz · 2021-11-29
> **Improved score, still insufficient experiments**
>
> I thank the authors for their response and revision. As the issue about the invariant/equivariant feature has been cleared up, I'll improve my score to a 5. Still, I think the experimental section is insufficient. In particular, the missing comparison to DeepSphere makes that I can not recommend acceptance. The argument that DeepSphere is in an entirely different category, because it is not exactly equivariant, seem unconvincing to me, as also the authors' method is not exactly equivariant, as tables 1 and 2 show. Additionally, even though Cobb (2021) may subsume previous architectures, a more thorough experimental evaluation would include more architectural variants. For example, this could include networks operating at higher frequencies (possibly at very high computational cost).

---

### Official Review · Reviewer_hrny · 2021-11-07

**Correctness:** 4
**Technical Novelty And Significance:** 2
**Empirical Novelty And Significance:** 2
**Recommendation:** 5
**Confidence:** 3

**Main Review:**

I thank the authors for this submission. It is a nicely written paper with a thorough explanation of the theory behind scattering networks and also some of the fundamentals such as Sec. 2.3. However, there are two fundamental problems:

- The technical contribution is not really high as most results come from previous works and the paper essentially contains a rather simple idea.
- The experiments miss a good application with a clear evaluation against previous work. I also could not see any quantitative results/comparisons on efficiency/ scalability, which is one of the main points of the paper.

I am thus more on the reject side, unfortunately.



**Summary Of The Paper:**

This paper proposes spherical scattering networks, which are scattering networks defined on the sphere and carry the nice properties of scattering networks such as invariance and stability.

**Summary Of The Review:**

This is a nicely written paper that contains an interesting idea. The lack of technical novelty and practical results lead me to rejection.

---

> ### Author Response · Authors · 2021-11-22
> **Response to Reviewer hrny**
>
> We thank the Reviewer for their comments.  We respond to each comment in turn below.  The Reviewer's original comments are italicized, while our responses are given in Roman font.  All revisions to the manuscript are highlighted in red.
>
> *The technical contribution is not really high as most results come from previous works and the paper essentially contains a rather simple idea.*
>
> The construction of scattering networks on the sphere is novel, although we appreciate that the construction follows from the direct analogy with the Euclidean construction of [Mallat (2012)](https://arxiv.org/abs/1101.2286) and theoretical results (i.e. Theorem 1 and 2) follow by minor modifications to the proofs of [Perlmutter et al. (2020)](https://arxiv.org/abs/1905.10448) for graphs.  Nevertheless, for the analysis of spherical data, which is widespread in many application domains, it is important that extensions to the sphere are made.
>
> In any case, the main contribution of the manuscript is the use of spherical scattering networks in combination with hybrid spherical CNNs to yield spherical machine learning techniques that are scalable to high-resolution, while also strictly rotationally equivariant.  There are no alternative methods that address this problem.  We believe a significant contribution of the manuscript is to solve this open problem in the field, even if the proposed approach results from a "simple idea" (which another Reviewer calls "elegant").
>
> *The experiments miss a good application with a clear evaluation against previous work. I also could not see any quantitative results/comparisons on efficiency/ scalability, which is one of the main points of the paper.*
>
> No previous spherical CCN techniques that are rotationally equivariant are able to scale to the resolutions considered in the manuscript.  Hence, we developed new benchmark experiments to test this scenario, where we are able to apply previous approaches on a downsampled version of the spherical images.  We compare to the generalized spherical CNNs developed by [Cobb et al. 2021](https://arxiv.org/abs/2010.11661), which achieve the start of the art and include spherical layers based on those of [Cohen et al. 2018](https://arxiv.org/abs/1801.10130), [Esteves et al. 2018](https://arxiv.org/abs/1711.06721), [2020](https://arxiv.org/abs/1711.06721), and [Kondor et al. 2018](https://arxiv.org/abs/1806.09231).
>
> Computational times were not previously given since application of the spherical scattering network is independent of training and is performed as a data-preprocessing step.  Execution and training of the subsequent hybrid spherical CNN that follows the scattering network is then the same as before when running at low resolution.  Nevertheless, while computation time of the scattering network does not impact training time, we agree readers may be interested in the the general time required to evaluate spherical scattering networks for the precompution.  This is approximately five seconds for the highest MNIST resolution considered ($L=256$) and a little under two minutes for the CMB resolution considered ($L=1024$).  However, we have *not* implemented a parallelized version of the spherical scattering transform: the current implementation runs sequentially on a single CPU core.  As mentioned, the scattering network evaluation is trivially parallelized in principle, hence there is great scope for reducing computation time.  We have revised the manuscript to quote these computation times and provide a brief related discussion.

---

### Decision · Program_Chairs · 2022-01-20

**Decision:**

Accept (Poster)

**Comment:**

The submission develops a rotationally equivariant scattering transform on the sphere.  Many developments in deep learning make use of spherical representations, and the development of a rotationally equivariant scattering transform is an important if not unexpected development.  The reviews are split with half of the reviewers believing it to be slightly above the threshold for acceptance, and half believe it to be slightly below the threshold for acceptance.  In the papers favor, it solves an important case of the scattering transform framework, which has been demonstrated to be important in diverse machine learning applications such as learning with small data sets, differentially private learning, and network initialization.  As such, continued fundamental development in this area is valuable, especially in the context of representation learning, the focus of ICLR.